**∂ | Open Peer Review** | Host-Microbial Interactions | Research Article

# SARS-CoV-2 infection is associated with intestinal permeability, systemic inflammation, and microbial dysbiosis in hospitalized patients

Christopher M. Basting,[1] Robert Langat,[1] Courtney A. Broedlow,[1] Candace R. Guerrero,[1,2,3] Tyler D. Bold,[4] Melisa Bailey,[1] Adrian Velez,[1] Ty Schroeder,[1] Jonah Short-Miller,[1] Ross Cromarty,[5] Zachary J. Mayer,[2,3] Peter J. Southern,[6] Timothy W. Schacker,[4] Sandra E. Safo,[7] Carolyn T. Bramante,[4] Christopher J. Tignanelli,[1] Luca Schifanella,[1,8] Nichole R. Klatt[1]

**ABSTRACT** Coronavirus disease 2019 (COVID-19) and its associated severity have been linked to uncontrolled inflammation and may be associated with changes in the microbiome of mucosal sites including the gastrointestinal tract and oral cavity. These sites play an important role in host–microbe homeostasis, and disruption of epithelial barrier integrity during COVID-19 may potentially lead to exacerbated inflammation and immune dysfunction. Outcomes in COVID-19 are highly disparate, ranging from asymptomatic to fatal, and the impact of microbial dysbiosis on disease severity is unclear. Here, we obtained plasma, rectal swabs, oropharyngeal swabs, and nasal swabs from 86 patients hospitalized with COVID-19 and 12 healthy volunteers. We performed 16S rRNA sequencing to characterize the microbial communities in the mucosal swabs and measured concentrations of circulating cytokines, markers of gut barrier integrity, and fatty acids in the plasma samples. We compared these plasma concentrations and microbiomes between healthy volunteers and COVID-19 patients, some of whom had unfortunately died by the end of the study enrollment, and performed a correlation analysis between plasma variables and bacterial abundances. Rectal swabs of COVID-19 patients had reduced abundances of several commensal bacteria including *Faecalibacterium prausnitzii* and an increased abundance of the opportunistic pathogens *Eggerthella lenta* and *Hungatella hathewayi*. Furthermore, the oral pathogen *Scardovia wiggsiae* was more abundant in the oropharyngeal swabs of COVID-19 patients who died. The abundance of both *H. hathewayi* and *S. wiggsiae* correlated with circulating inflammatory markers including IL-6, highlighting the possible role of the microbiome in COVID-19 severity and providing potential therapeutic targets for managing COVID-19.

**IMPORTANCE** Outcomes in coronavirus disease 2019 (COVID-19) are highly disparate and are associated with uncontrolled inflammation; however, the individual factors that lead to this uncontrolled inflammation are not fully understood. Here, we report that severe COVID-19 is associated with systemic inflammation, microbial translocation, and microbial dysbiosis. The rectal and oropharyngeal microbiomes of COVID-19 patients were characterized by a decreased abundance of commensal bacteria and an increased abundance of opportunistic pathogens, which positively correlated with markers of inflammation and microbial translocation. These microbial perturbations may, therefore, contribute to disease severity in COVID-19 and highlight the potential for microbiome-based interventions in improving COVID-19 outcomes.

**KEYWORDS** COVID-19, human microbiome, gut microbiome, host–pathogen interactions

Address correspondence to Nichole R. Klatt, klat0037@umn.edu.

The authors declare no conflict of interest.

Severe acute respiratory syndrome coronavirus 2 (SARS-CoV-2), the virus responsible for the coronavirus disease 2019 (COVID-19), is known for its highly disparate outcomes, ranging from asymptomatic to severe illness, and death. However, the factors that determine various outcomes and susceptibility to severe versus mild illness among individuals are still not well understood (1, 2). Numerous studies have reported that patients who are older (≥60), are male, and have comorbidities including obesity, diabetes, and cardiovascular disease are at increased risk of developing severe COVID-19 and related mortality (3–6), though the mechanisms that drive this increased risk are poorly understood (7).

One of the first lines of defense against viral infections is an effective cytokine response that induces an appropriate inflammatory response. However, this can be a double-edged sword; COVID-19 can be accompanied by an aggressive inflammatory response and overwhelming release of cytokines referred to as a "cytokine storm" (8–11). Cytokine storm, a term for overt or unregulated inflammation, may ultimately result in multi-organ failure and death (10). This unregulated inflammation can cause severe clinical complications including acute respiratory distress syndrome, one of the leading causes of death in patients with COVID-19, likely induced by an excessive immune response rather than the SARS-CoV-2 virus itself (12). Inflammation in COVID-19 is predictive of mortality and is associated with increased plasma concentrations of several cytokines including IL-2, IL-6, IL-7, IL-10, and TNF-α (13). In addition, tissue analysis from the lungs of deceased COVID-19 patients demonstrates that inflammation-induced apoptosis, necroptosis, and pyroptosis, and not solely viral replication, is associated with death (14). The mechanisms that contribute to the development of this overt inflammation are not fully understood, though an imbalance in host pro-inflammatory and anti-inflammatory cytokines is a common feature (9). Furthermore, the inflammation induced by a lung infection can lead to intestinal permeability, resulting in increased microbial translocation of gut microbes and microbial products such as lipopolysaccharide (LPS), similar to observed drivers of mortality in HIV (15–18). The net effect of microbial translocation can enhance systemic inflammation and lung injury, driving severity during respiratory-related diseases (19, 20). Additionally, SARS-CoV-2 can directly infect intestinal enterocytes (21), leading to the breakdown of the epithelial barrier (22) and possibly contributing to microbial translocation. Therefore, the role of the gut microbiome in systemic inflammation and cytokine storm in COVID-19, as well as other mucosal epithelial sites in the upper airways such as the oropharynx and nares, is an important area of investigation.

The human gastrointestinal tract harbors trillions of microorganisms that form an ecological community known as gut microbiota, which, when altered, results in dysbiosis and has been associated with various human diseases (23). The gut microbiome is consistently associated with disease severity, immunological dysfunction, and long-term outcomes in viral infections and has also been demonstrated in the context of COVID-19 (24, 25). Several studies have reported that COVID-19 is associated with drastic alterations of the normal intestinal flora, even when removing the confounding effect of antibiotics (24, 26, 27). For instance, COVID-19 is associated with a decrease in critical butyrate-producing bacteria, including *Faecalibacterium* and *Roseburia* (28–30). The genus *Roseburia* contributes to mucosal integrity and colonic motility and exerts significant anti-inflammatory effects by modulating IL-10 production (31). Likewise, the species *Faecalibacterium prausnitzii* is a valuable gut symbiont with recognized anti-inflammatory effects in patients with inflammatory bowel disease via the inhibition of the NF-κB pathway (32). Conversely, opportunistic pathogens, which are commonly overrepresented in the COVID-19-related gut microbiome, such as *Bacteroides dorei*, have been associated with increased levels of IL-6 and IL-8 (29). Pro-inflammatory cytokines such as IL-6 have further been linked to increased levels of LPS-binding protein (LBP), β-glucan, and zonulin, suggesting that microbial translocation and gut barrier permeability increase systemic inflammation (28–30, 33). Microbial dysbiosis in the gut microbiome has been shown to correlate with higher levels of inflammatory cytokines in

patients with COVID-19, suggesting that the gut microbiome is involved in the magnitude of COVID-19 severity, possibly via the modulation of the host immune response (34). Furthermore, alterations in the oral microbiome have been seen in COVID-19 patients including a reduced diversity, especially in older patients (35), and bacteria associated with periodontal disease have been shown to stimulate IL-6 and TNF-α production by immune cells (36). Thus, the oral microbiome may also be a source of inflammation and cytokine production that contributes to COVID-19 severity. However, the role of the microbiome in COVID-19 has not been fully elucidated in its ability to contribute to disease severity.

Short-chain fatty acids (SCFAs) are primarily produced by the anaerobic fermentation of nondigestible carbohydrates by gut bacteria while medium-chain fatty acids (MCFAs) are largely derived from dietary sources (37, 38). The most abundantly produced SCFAs are acetate, butyrate, and propionate (39), each of which has its own effects. Butyrate is the main energy source for colonocytes (40) and has demonstrated importance in maintaining intestinal epithelial barrier integrity (41). Previous research shows that both butyrate and propionate inhibit the expression of the pro-inflammatory cytokines IL-6 and TNF-α as well as modulate the differentiation and activation of immune cells by signaling through G protein-coupled receptors (42, 43). Propionate has also been shown to have anti-inflammatory properties by decreasing IL-6 release in colon organ cultures (44) and strengthening tight junction barriers in intestinal epithelial cells, thereby reducing intestinal permeability (45). COVID-19 has previously demonstrated an association with reduced SCFA-producing bacteria in the gut such as *Roseburia* (46), which may result in reduced SCFA production and a loss of their potential health benefits. Furthermore, dietary changes during hospitalization may also result in changes in MCFA concentrations. Therefore, evaluating the concentrations of circulating fatty acids in COVID-19 may inform important biological changes during disease progression.

Here, we investigated cytokine profiles, circulating fatty acids, markers of gut barrier function, and the microbiota of nasal, oropharyngeal, and rectal swabs in hospitalized COVID-19 patients, of whom some survived while others unfortunately died from COVID-19 complications. Several noninfected, healthy volunteers were also included to better understand the relationships between the microbiome of different mucosal sites, clinical features, and systemic host immune responses.

## RESULTS

### Characteristics of the study participants and study overview

We divided the hospitalized SARS-CoV-2-positive individuals into two severity groupings based on whether they had survived ($n = 69$) or died ($n = 17$) by the end of study enrollment, with the deceased group being defined here as having a more severe illness. The average age of patients who died was significantly higher than those who survived ($P = 0.003$). Deceased patients also had longer hospitalizations ($P = 0.003$) and were predominantly male (76%) compared to those who survived (50.7%), though this difference was not statistically significant ($P = 0.377$). Patients who died also had a higher prevalence of several comorbidities, including cardiovascular disease, hypertension, and diabetes; however, a prior history of cancer was the only statistically significant difference ($P = 0.003$). No significant differences were observed between the two groups regarding neutrophil counts, lactate dehydrogenase (LDH), aspartate transaminase (AST), alanine transaminase (ALT), and creatinine kinase levels, though deceased patients had significantly lower albumin levels compared to those who survived ($P = 0.012$). Deceased COVID-19 patients also had significantly higher SARS-CoV-2 viral loads in the oropharyngeal swabs compared to those who survived ($P = 0.030$). SARS-CoV-2 viral loads in the nasal swabs were higher in the deceased COVID-19 patients as well, though this difference was not statistically significant ($P = 0.202$). Clinical characteristics, hospital lab results, and viral load measurements for the COVID-19 patients are shown in Table 1; Fig. S1. As previously reported (3), patients who died were primarily older, were male, and had more pre-existing conditions than those who survived. They also

**TABLE 1** Clinical characteristics of COVID-19 patients[a]

| | Survived (n = 69) | Deceased (n = 17) | %missing | P-value |
|---|---|---|---|---|
| Age [years, mean (SD)] | 59.26 (15.41) | 71.82 (10.55) | 0 | 0.003[b] |
| Gender | | | 0 | 0.377[d] |
| Female [n (%)] | 34 (49.3) | 4 (23.5) | | |
| Male [n (%)] | 35 (50.7) | 13 (76.5) | | |
| Length of hospitalization [days, median (IQR)] | 6.00 (4.00, 12.00) | 13.00 (8.00, 19.00) | 0 | 0.003[c] |
| Length on ventilator devices [median (IQR)] | 2.00 (0.00, 5.00) | 9.00 (0.00, 14.00) | 0 | 0.101 |
| ICU visit during admit [n (%)] | 43 (62.3) | 13 (76.5) | 0 | 0.625[d] |
| Length of ICU stay [days, median (IQR)] | 2.00 (0.00, 5.00) | 5.00 (0.00, 13.00) | 0 | 0.142[c] |
| Hospital lab results | | | | |
| Albumin [g/dL, mean (SD)] | 2.88 (0.45) | 2.41 (0.44) | 36 | 0.012[b] |
| AST [U/L, median (IQR)] | 35.00 (23.00, 76.00) | 34.14 (28.80, 41.00) | 37.2 | 0.967[c] |
| ALT [U/L, median (IQR)] | 45.25 (26.63, 86.25) | 48.50 (31.60, 67.00) | 38.4 | 0.967[c] |
| LDH [U/L, mean (SD)] | 332.10 (112.31) | 381.10 (110.89) | 20.9 | 0.269[b] |
| Neutrophil count [thousand/µL, mean (SD)] | 6.70 (3.04) | 8.12 (3.50) | 7 | 0.266[b] |
| Creatine kinase [U/L, median (IQR)] | 171.67 (30.00, 298.00) | 51.33 (45.00, 76.75) | 67.4 | 0.493[c] |
| SARS-CoV-2 N1 viral load (copies/µL) | | | | |
| Nasal swab [median (IQR)] | 3.22 (0.22, 58.41) | 44.82 (1.49, 2,039.58) | 30.2 | 0.202[c] |
| Oropharyngeal swab [median (IQR)] | 0.84 (0.00, 23.35) | 118.07 (9.86, 371.51) | 24.4 | 0.030[c] |
| Rectal swab [median (IQR)] | 0.03 (0.00, 0.66) | 0.00 (0.00, 0.00) | 76.7 | NA |
| Comorbidities | | | | |
| Oncologic disease [n (%)] | 0 (0 %) | 5 (29.4%) | 0 | 0.003[e] |
| Coronary artery disease [n (%)] | 6 (8.7%) | 6 (35.3%) | 0 | 0.109[d] |
| Heart failure [n (%)] | 5 (7.2%) | 4 (23.5%) | 0 | 0.357[e] |
| Cardiovascular disease [n (%)] | 24 (34.8%) | 9 (52.9%) | 0 | 0.581[d] |
| Hypertension [n (%)] | 24 (34.8%) | 9 (52.9%) | 0 | 0.581[d] |
| Hematologic disease [n (%)] | 15 (21.7%) | 7 (41.2%) | 0 | 0.546[d] |
| Diabetes [n (%)] | 15 (21.7%) | 6 (35.3%) | 0 | 0.625[d] |
| Hyperlipidemia [n (%)] | 18 (26.1%) | 7 (41.2%) | 0 | 0.625[d] |
| Obesity [n (%)] | 14 (20.3%) | 5 (29.4%) | 0 | 0.724[d] |
| Nephrologic disease [n (%)] | 21 (30.4%) | 7 (41.2%) | 0 | 0.724[d] |
| Metabolic disease—non-diabetes [n (%)] | 29 (42%) | 9 (52.9%) | 0 | 0.724[d] |
| Rheumatic disease [n (%)] | 24 (34.8%) | 7 (41.2%) | 0 | 0.893[d] |
| Pulmonary disease [n (%)] | 20 (29%) | 5 (29.4%) | 0 | 1.000[d] |

[a]Continuous variables are reported as the mean and standard deviation (SD) for normally distributed and median and interquartile range (IQR) for nonnormally distributed variables.
[b]Two-sided Welch t-test.
[c]Two-sided Wilcox test.
[d]Chi-squared test.
[e]Fisher exact test.

tended to have longer hospitalizations, higher SARS-CoV-2 viral loads, and lower albumin concentrations.

## Severe COVID-19 is associated with high levels of pro-inflammatory cytokines and markers of intestinal permeability

To assess the systemic inflammatory response of severe COVID-19, we measured the concentrations of numerous pro- and anti-inflammatory cytokines in the plasma of hospitalized COVID-19 patients and several healthy controls, summarized in Table 2; Fig. 1. IL-6 plasma concentrations were highest in COVID-19 patients who died followed by those who survived and healthy controls {Med: 117 pg/mL [interquartile range (IQR): 58.6–268.5] vs 5.96 pg/mL (IQR: 3.14–25.82) vs 1.42 pg/mL (IQR: 1.29–1.67), respectively, Kruskal–Wallis, $P < 0.05$} with significant pairwise differences between all groups (Fig. 1A). Similarly, IL-2 was also highest in deceased COVID-19 patients followed by patients who survived and healthy controls [Med: 0.79 pg/mL (IQR: 0.39–0.97) vs 0.29 pg/mL (IQR:

**TABLE 2** Plasma biomarker concentrations in all enrolled patients[a]

| | Healthy (*n* = 5) | Survived (*n* = 38) | Deceased (*n* = 10) | %missing | *P*-value |
|---|---|---|---|---|---|
| **Cytokines (pg/mL)** | | | | | |
| IFN-γ | 0.86 (0.67, 0.92) | 0.78 (0.49, 1.37) | 2.37 (0.84, 5.69) | 0 | 0.320 |
| IL-10 | 2.02 (1.99, 2.18) | 4.63 (2.91, 8.30) | 9.56 (5.70,13.42) | 0 | 0.002 |
| IL-12p70 | 1.28 (1.20, 1.52) | 0.88 (0.64, 1.28) | 0.86 (0.50, 1.02) | 0 | 0.233 |
| IL-18 | 183.00 (166.00, 293.00) | 441.00 (278.00, 613.75) | 397.00 (203.25, 931.00) | 0 | 0.107 |
| IL-1β | 0.62 (0.62, 0.70) | 0.44 (0.26, 0.71) | 0.32 (0.21, 0.68) | 0 | 0.687 |
| IL-2 | 0.23 (0.22, 0.27) | 0.29 (0.19, 0.44) | 0.79 (0.39, 0.97) | 0 | 0.046 |
| IL-6 | 1.42 (1.29, 1.67) | 5.96 (3.14, 25.82) | 117.00 (58.60, 268.50) | 0 | <0.001 |
| TNF-α | 6.98 (6.23, 7.20) | 11.55 (8.06, 14.55) | 16.15 (10.77, 26.70) | 0 | 0.064 |
| IL-23 | 20.40 (11.52, 27.50) | 40.57 (23.45, 68.72) | 28.14 (22.16, 51.12) | 0 | 0.233 |
| IL-17A | 0.89 (0.55, 1.01) | 1.25 (0.90, 1.57) | 0.94 (0.68, 1.25) | 0 | 0.267 |
| IL-8 | 0.12 (0.12, 4.84) | 0.68 (0.25, 1.88) | 1.64 (0.63, 7.44) | 0 | 0.196 |
| **Gut barrier markers (ng/mL)** | | | | | |
| sCD14 | 1,772.46 (1,633.74, 1,965.35) | 2,128.58 (1,661.82, 3,071.70) | 2,922.96 (2,180.24, 3,705.77) | 0 | 0.233 |
| Zonulin | 5.35 (4.16, 8.95) | 10.53 (7.57, 17.45) | 10.95 (6.63, 17.56) | 0 | 0.196 |
| I-FABP | 1.37 (1.04, 1.44) | 1.27 (0.78, 1.76) | 0.95 (0.68, 1.15) | 5.6 | 0.548 |
| LBP | 27,426.06 (27,113.30, 28,709.94) | 36,405.90 (28,813.66, 54,620.61) | 79,024.78 (60,555.45, 97,675.48) | 0 | 0.007 |
| **Fatty acids (ng/μL)** | | | | | |
| Propionic acid | 8.12 (7.95, 8.40) | 5.24 (3.99, 5.88) | 5.79 (4.41, 6.19) | 1.9 | 0.015 |
| Butyric acid | 0.41 (0.38, 0.50) | 0.40 (0.27, 0.54) | 0.44 (0.35, 0.52) | 1.9 | 0.834 |
| Nonanoic acid | 0.83 (0.68, 0.97) | 1.06 (0.92, 1.42) | 1.10 (0.93, 1.33) | 1.9 | 0.320 |
| Decanoic acid | 1.69 (1.50, 2.16) | 0.21 (0.09, 0.39) | 3.59 (0.90, 7.80) | 1.9 | 0.002 |
| Isovaleric acid | 0.15 (0.13, 0.18) | 0.27 (0.20, 0.33) | 0.25 (0.21, 0.32) | 1.9 | 0.158 |

[a]Data are shown as the median (IQR); all *P*-values are determined by Kruskal–Wallis tests.

0.19–0.44) vs 0.23 pg/mL (IQR: 0.22–0.27), respectively, Kruskal–Wallis, *P* < 0.05], though the only significant pairwise comparisons were between the deceased and healthy (*P* = 0.035) and deceased and survived (*P* = 0.018) groups (Fig. 1B). TNF-α concentrations also appeared to increase with disease severity, with the highest concentrations seen in deceased patients, followed by those who survived and healthy controls; however, this difference was not statistically significant [Med: 16.15 pg/mL (IQR: 10.77–26.70), 11.55 pg/mL (IQR: 8.06–14.55), and 6.98 pg/mL (IQR: 6.23–7.20), respectively, Kruskal–Wallis, *P* = 0.064] (Fig. 1K). For anti-inflammatory cytokines, IL-10 was significantly higher in deceased COVID-19 patients compared to patients who survived and healthy controls [Med: 9.56 pg/mL (IQR: 5.23–22.25) vs 4.56 pg/mL (IQR: 2.86–8.17) vs 2.01 pg/mL (IQR: 1.41–2.71), respectively, Kruskal–Wallis, *P* < 0.05] with significant pairwise differences between all groups (Fig. 1C). The concentrations of several other cytokines also appeared to be increased in the hospitalized COVID-19 patients including IL-8, IL-18, and IFN-γ; however, none of these were statistically significant after adjusting for multiple comparisons (Kruskal–Wallis, *P* > 0.05).

Next, we asked whether severe COVID-19 is associated with differences in markers of intestinal barrier integrity and microbial translocation, given the potential for mucosal dysfunction driving systemic inflammation (49, 50). To this end, we measured the plasma concentrations of four key markers of intestinal permeability, intestinal fatty acid-binding protein (I-FABP), LBP, soluble CD14 (sCD14), and zonulin, which are summarized in Table 2; Fig. 2. The only marker of intestinal permeability that was statistically significant between these groups was LBP, where we observed considerably higher concentrations in COVID-19 patients who died compared to both healthy controls and patients who survived (*P* = 0.003 and *P* = 0.005, respectively; Fig. 2C). LBP is indicative of the amount of bacterial LPS present (51, 52), which has previously been shown to be increased in severe cases of COVID-19 (53). Interestingly, the SARS-CoV-2 spike protein has also been shown to be able to bind to LPS and boost pro-inflammatory activity (54). Additional markers

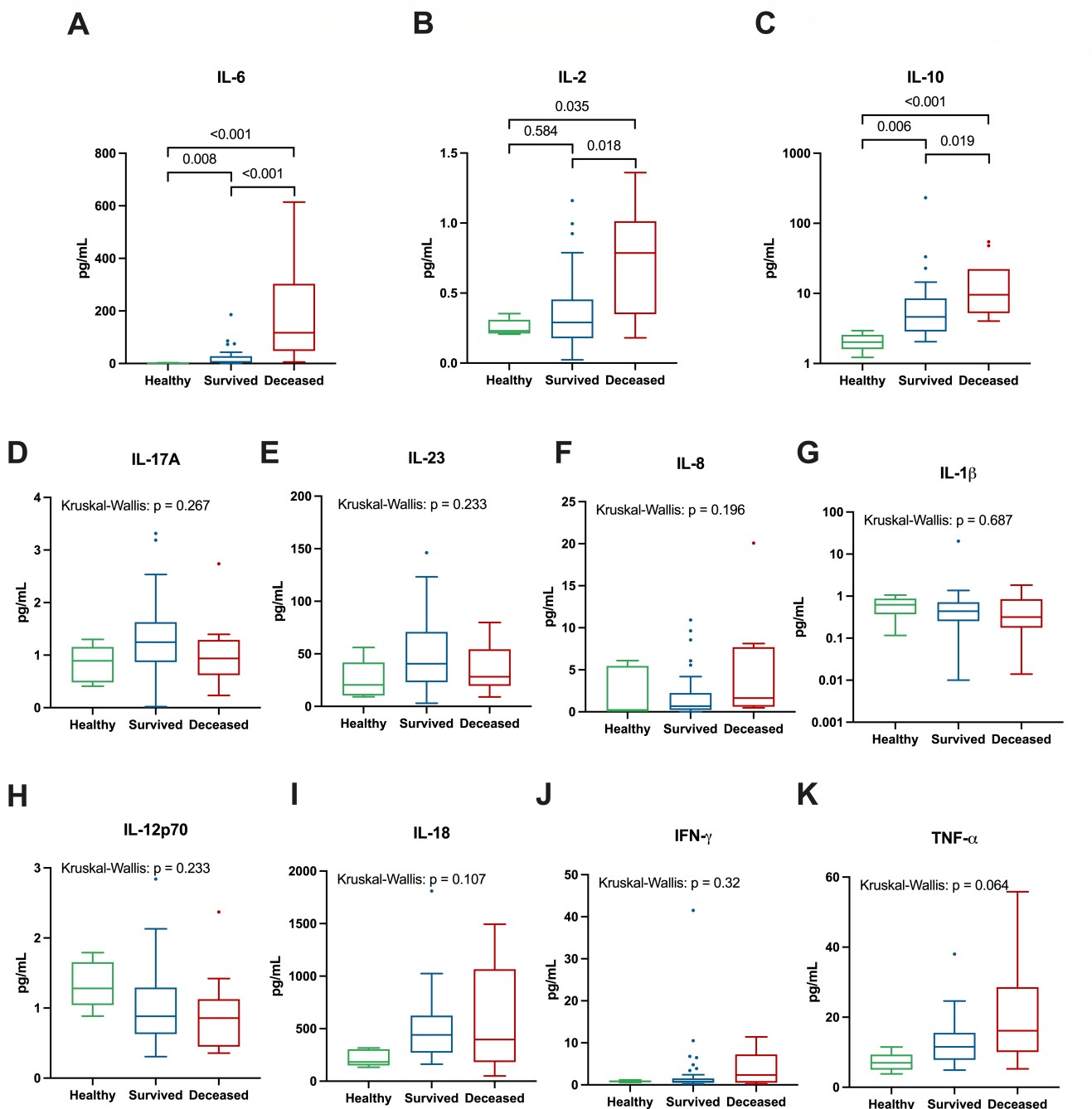

**FIG 1** Plasma concentrations of cytokines in COVID-19 patients and healthy controls. The plasma concentrations of IL-6 (A), IL-2 (B), IL-10 (C), IL-17A (D), IL-23 (E), IL-8 (F), IL-1β (G), IL-12p70 (H), IL-18 (I), IFN-γ (J), and TNF-α (K) from hospitalized COVID-19 patients and controls. Tukey's box shows the IQR; middle line, median; vertical lines, adjacent values (1st −1.5 IQR; 3rd quartile +1.5 IQR). Kruskal–Wallis tests were performed for all analytes but only followed by pairwise Dunn tests if the false discovery rate (FDR)-adjusted *P*-value was <0.05; thus, all plots showing pairwise comparisons had significant Kruskal–Wallis tests. Pairwise Dunn tests within each cytokine were also FDR-adjusted using the Benjamini–Hochberg method (47, 48). *P*-values <0.05 were considered significant.

of intestinal permeability, zonulin and sCD14, were both elevated in COVID-19 patients compared to healthy controls; however, these differences were not significant (Kruskal–Wallis, *P* > 0.05; Fig. 2B through D). These results overall suggest that severe COVID-19 is associated with increased concentrations of pro-inflammatory and anti-inflammatory cytokines as well as markers of intestinal permeability and microbial translocation and that several of these markers increase with disease severity.

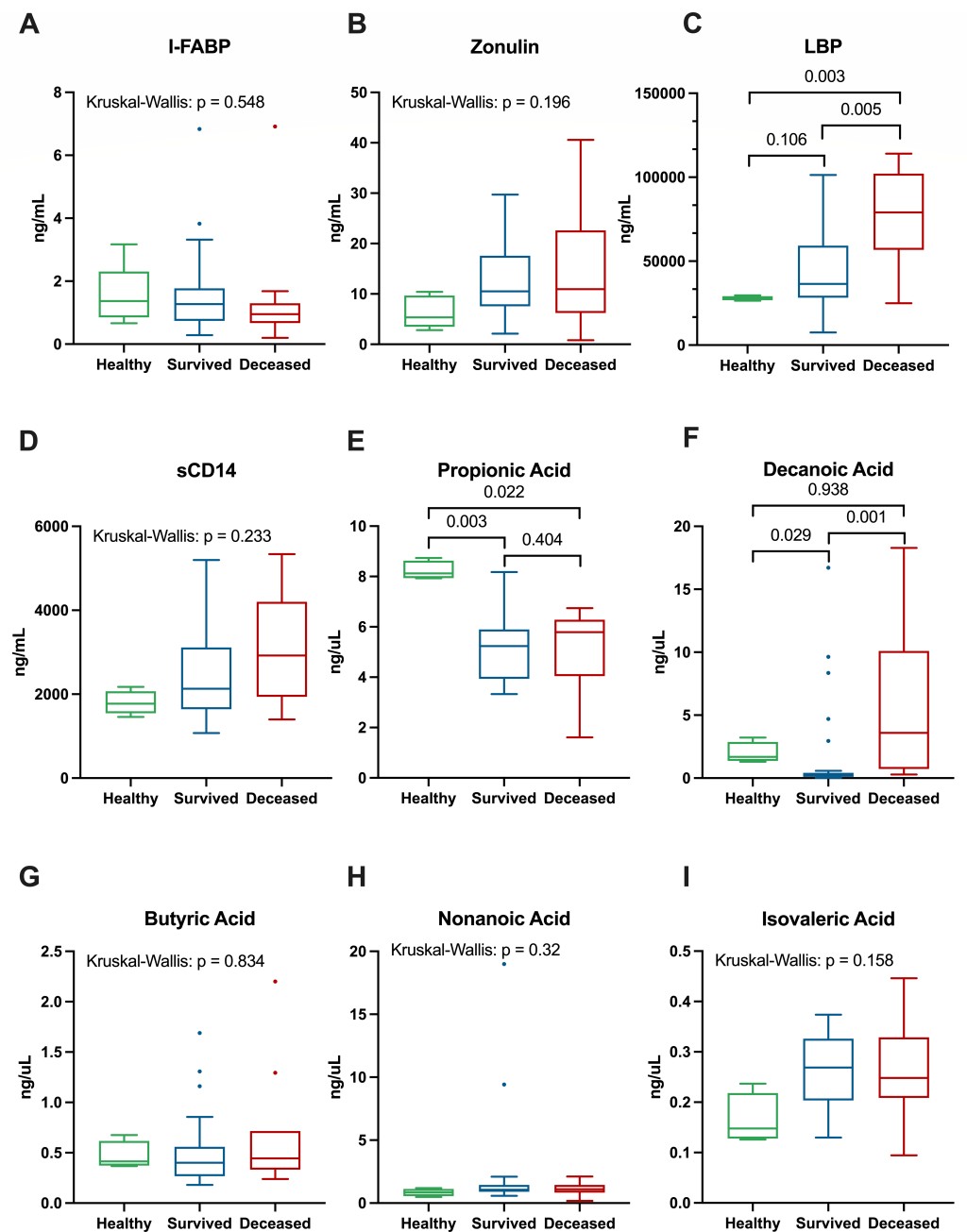

**FIG 2** Plasma concentrations of gut barrier damage markers and circulating fatty acids in COVID-19 patients and healthy controls. Plasma concentrations for I-FABP (A), zonulin (B), LBP (C), and sCD14 (D) measured by enzyme-linked immunosorbent assay (ELISA) and circulating fatty acids propionic acid (E), decanoic acid (F), butyric acid (G), nonanoic acid (H), and isovaleric acid (I) measured by liquid chromotagraphy-tandem mass spectrometry ( LC-MS/MS). Comparisons between groups were performed by Kruskal–Wallis tests for all analytes, but only followed by Dunn *post hoc* tests if adjusted *P*-values were below 0.05; thus, all plots showing pairwise comparisons had significant Kruskal–Wallis tests. Pairwise Dunn tests within each variable were also FDR-adjusted using the Benjamini–Hochberg method (48). *P*-values <0.05 were considered significant.

## Alterations in circulating fatty acids of severe COVID-19

To assess whether severe COVID-19 is associated with changes in circulating fatty acids, which are critical for maintaining mucosal integrity and regulating immunity (55), we measured several SCFAs and MCFAs in plasma including propionate, butyrate, isovaleric acid, nonanoic acid, and decanoic acid (Fig. 2; Table 2). Propionate was reduced in both COVID-19 patients who survived and those who died compared to healthy controls (*P*

= 0.003 and $P$ = 0.022, respectively). There was no difference in the concentrations of butyrate or isovaleric acid between any of the groups (Kruskal–Wallis, $P > 0.05$). For the MCFAs, decanoic acid was significantly higher in COVID-19 patients who died compared to those who survived ($P = 0.001$); however, there was no difference between healthy controls ($P = 0.938$). Furthermore, decanoic acid was increased in the healthy controls compared to COVID-19 patients who survived ($P = 0.029$). There was no difference in nonanoic acid concentrations between any of the groups (Kruskal–Wallis, $P > 0.05$). Overall, these results suggest that severe COVID-19 is associated with several changes in the circulating concentrations of both SCFAs and MCFAs, which may underlie the damage in the gut leading to microbial translocation.

## Relationships between cytokines, viral loads, and gut barrier damage in severe COVID-19

To examine the relationships between cytokines, SCFAs, markers of gut barrier damage, hospital lab results, and SARS-CoV-2 viral loads, we performed a Spearman correlation analysis (Fig. 3). We observed strong correlations between several pro-inflammatory cytokines, such as IL-6, TNF-α, IL-2, and IFN-γ. For example, TNF-α positively correlated with IL-6 ($r = 0.61$, $P = 3.55e{-}5$) and IFN-γ ($r = 0.63$, $P = 1.41e{-}5$). We also observed a strong correlation between several pro-inflammatory and anti-inflammatory cytokines, such as IL-6 and IL-10 ($r = 0.63$, $P = 1.41e{-}5$). Many pro-inflammatory cytokines were also strongly positively correlated with markers of gut barrier damage, such as IL-6 and LBP ($r = 0.64$, $P = 1.28e{-}5$), IFN-γ and sCD14 ($r = 0.68$, $P = 2.69e{-}6$), and IL-18 and zonulin ($r = 0.83$, $P = 2.09e{-}12$). We observed limited correlations between SCFAs and other biomarkers; however, both butyrate and isovaleric acid were negatively correlated with IL-12p70 ($r = -0.43$, $P = 0.015$ and $r = -0.51$, $P = 0.001$, respectively). Albumin concentrations were negatively correlated with several pro-inflammatory cytokines including IL-6 ($r = -0.56$, $P = 0.009$) and IL-2 ($r = -0.54$, $P = 0.012$) as well as LBP ($r = -0.54$, $P = 0.012$). Lastly, we observed positive correlations between oropharyngeal SARS-CoV-2 viral loads and IL-10 ($r = 0.64$, $P = 2.23e{-}4$), IL-6 ($r = 0.55$, $P = 0.004$), and TNF-α ($r = 0.49$, $P = 0.013$) as well as an expected positive correlation with nasal SARS-CoV-2 viral loads ($r = 0.57$, $P \leq 0.001$). These correlations highlight the relationships between SARS-CoV-2 infection with systemic host immune responses and inflammation and intestinal barrier dysfunction. Furthermore, these associations provide a potentially novel mechanism by which systemic inflammation caused by SARS-CoV-2 infection leads to intestinal permeability and increased microbial translocation, which further drives systemic inflammation in a vicious feedback cycle.

## Identification of features that characterize severe COVID-19

To identify the most important variables in discriminating between patients who died or survived or healthy controls, we performed a supervised clustering analysis with sparse partial least squares-discriminant analysis (sPLS-DA) using the MixOmics R package (56) (Fig. 4). We saw moderate overlap between the deceased and survived groups, with less overlap seen with the healthy control group. The majority of separation between groups occurs in component 1 of the sPLS-DA model, with the variables that contribute the most mainly being inflammatory cytokines (primarily IL-6, IL-2, and TNF-α), markers of gut barrier damage (LBP and sCD14), decanoic acid, and the nasal swab viral load. These variables are elevated in patients who died and contributed to their separation compared to patients who survived or healthy controls. Additionally, albumin, IL-23, and IL-17A contributed to the separation of the survived and healthy groups along component 1. Component 2 of the sPLS-DA model largely separated the healthy controls from the COVID-19 patients. The variable that contributed the most to this component was propionic acid, which was elevated in the healthy controls. Overall, this supervised analysis shows that pro-inflammatory cytokines, several markers of gut barrier damage, and SCFAs are the most important variables for characterizing severe COVID-19 within this data set.

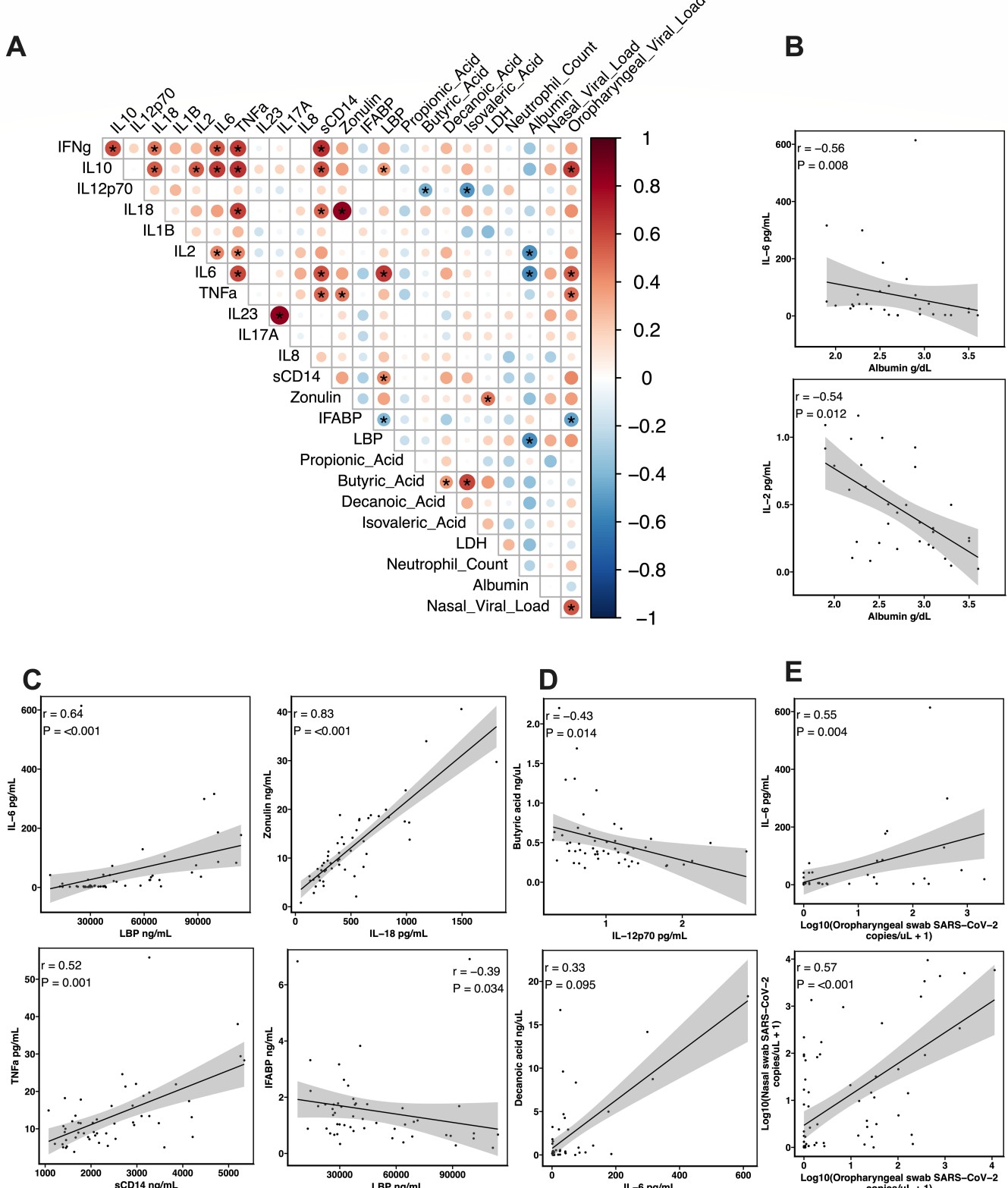

**FIG 3** Correlations between cytokines, markers of gut barrier damage, fatty acids, hospital lab results, and SARS-CoV-2 viral loads. Spearman correlation plot; asterisks denote FDR-adjusted *P*-values <0.05 (A). Representative correlations of interest for hospital labs are shown in (B), gut barrier damage in (C), short-chain fatty acids in (D), and SARS-CoV-2 viral loads in (E). Lines represent the simple linear regression, and shaded areas represent the 95% confidence intervals.

**A**

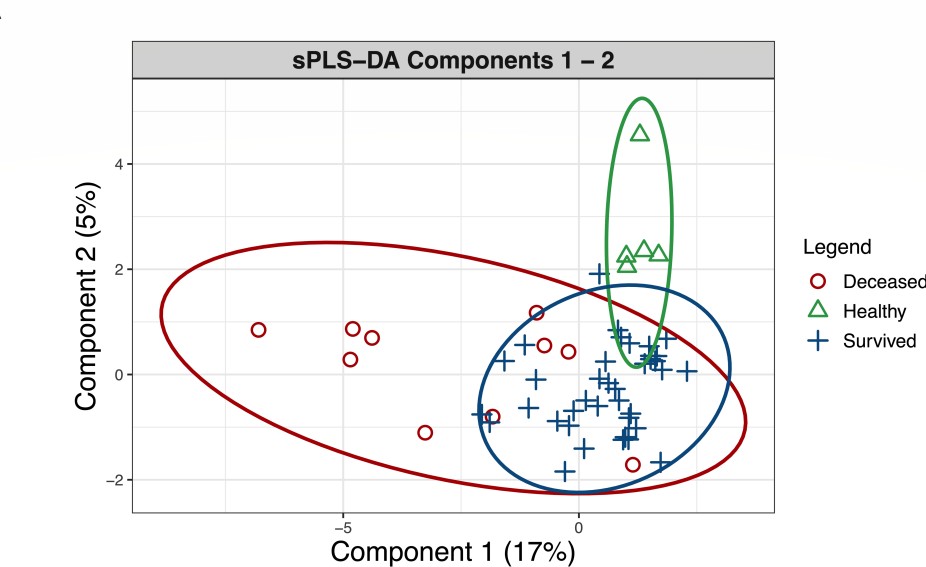

**B**

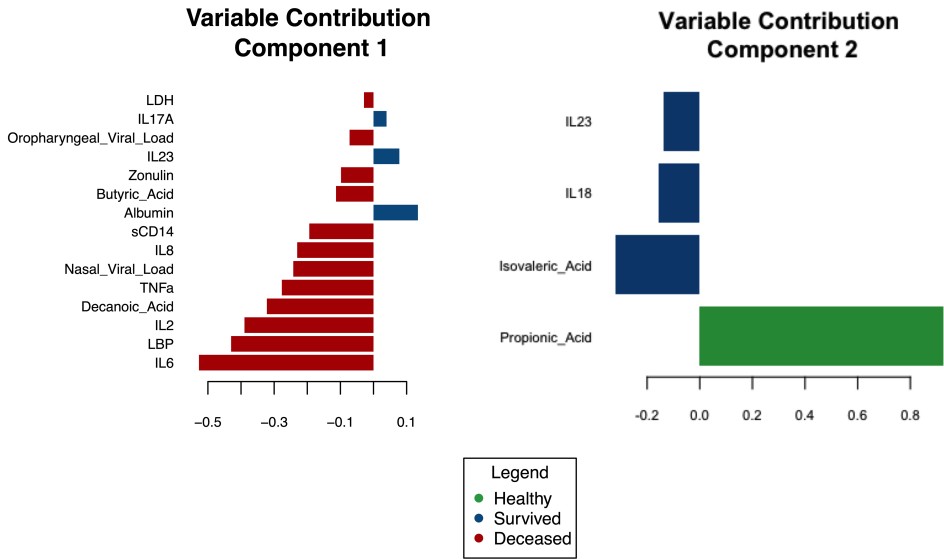

**FIG 4** Identifying biomarkers most important in separating COVID-19 patients and healthy controls. sPLS-DA of plasma biomarkers, hospital labs, and swab SARS-CoV-2 viral loads from oropharyngeal and nasal swabs used to identify the variables most important in discriminating patients who died or survived or healthy control. The ordination of the samples is shown in component 1 and component 2 and colored by their group (A). Variables contributing to those components are shown below (B) and indicate each variable's loading value and direction. Each variable is colored by the group that had the highest median value.

## Changes in the nasal, oropharyngeal, and rectal microbiomes of severe COVID-19 patients

Given the profound impact of the microbiome on mucosal health, we next investigated changes in the microbiome during severe COVID-19 infection. We performed 16S rRNA sequencing of the V3V4 region in rectal, oropharyngeal, and nasal swabs collected from healthy controls and hospitalized COVID-19 patients who had either survived or died

by the end of study enrollment. Due to the limited number of rectal swabs collected from deceased patients ($n = 2$), we compared the rectal swabs of the healthy controls to all hospitalized patients. We observed reduced alpha diversity in the rectal swabs of the COVID-19 patients compared to the healthy controls ($P = 0.013$; Fig. 5A), as well as a lower alpha diversity in the oropharyngeal swabs of patients who died compared to healthy controls ($P = 0.003$; Fig. 5B). Alpha diversity was highest in the healthy controls of the oropharyngeal swabs and appeared to decline with COVID-19 severity. We did not observe any significant differences in the alpha diversity of the nasal swabs. Permutational multivariate analysis of variance (PERMANOVA) showed a significant difference in the overall microbial composition in the rectal swabs ($P = 0.007$, PERMANOVA; Fig. 5A), oropharyngeal swabs ($P = 0.003$, PERMANOVA; Fig. 5B), and nasal swabs ($P = 0.045$, PERMANOVA; Fig. 5C). Within the COVID-19 patients, the primary factor that contributed to the rectal swab composition was antibiotic usage prior to sample collection; however, its effect was not statistically significant ($P = 0.093$; Fig. 5A). Age was the primary factor explaining variance in the oropharyngeal swabs ($P = 0.003$; Fig. 5B), while COVID-19 grouping (survived/deceased) was the primary factor for the nasal swabs ($P = 0.025$; Fig. 5C). Furthermore, the within-group compositional dissimilarity, or beta-dispersion, of the rectal and oropharyngeal samples was significantly higher in the COVID-19 patients, which appeared to increase with disease severity, whereas the beta-dispersion of the nasal swabs was not significantly different between any groups (Fig. 5A through C).

## Enrichment of opportunistic pathogens and depletion of commensal bacteria in severe COVID-19

To further characterize the differences in the rectal, oropharyngeal, and nasal microbiomes of severe COVID-19 patients, we performed differential abundance testing at the phylum, genus, and species levels. We identified numerous differentially abundant taxa in the rectal swabs of healthy controls and the hospitalized COVID-19 patients (Fig. 6A). At the species level, there was an enrichment of *Eggerthella lenta* and *Hungatella hathewayi* in the COVID-19 patients and a depletion of *Faecalibacterium prausnitzii*, *Fusicatenibacter saccharivorans*, and *Dorea longicatena*. At the genus level, there was an enrichment of *Actinomyces*, *Clostridium innocuum* group, and *Eubacterium brachy* group in the COVID-19 patients and a depletion of several commensal bacteria including *Coprococcus*, *Dorea*, *Ruminococcus,* and *Roseburia*. Pairwise comparisons between COVID-19 patients who survived or died and healthy controls were performed for the nasal and oropharyngeal swabs (Fig. 6B). At the species level, the oropharyngeal swabs of COVID-19 patients who died were characterized by an enrichment of *Veillonella parvula* and *Scardovia wiggsiae* compared to both survivors and healthy controls and depletion of *Veillonella atypica*. At the genus level, *Lactobacillus, Howardella,* and *Scardovia* were enriched in the COVID-19 patients who died and *Fusobacterium* and *Leptotrichia* were depleted relative to healthy controls and patients who survived. At the phylum level, deceased patients had a significant depletion of Fusobateriota compared to healthy controls and patients who survived. Both COVID-19 patients who survived and died had a depletion of several taxa compared to healthy controls including *Haemophilus, Stomatobaculum, Catonella, Selenomonas, Lachnoanaerobaculum, TM7x, Candidatus Saccharimonas, Mogibacterium, Fusobacterium periodonticum,* and *Campylobacter concisus*. The only differentially abundant taxa observed in the nasal swabs was a depletion of *Cutibacterium* in COVID-19 patients who died compared to those who survived or healthy controls (Fig. 6B). Taken together, these alterations to the microbiome in COVID-19 patients may underlie the mucosal dysfunction observed.

## Abundance of opportunistic pathogens correlates with pro-inflammatory markers, tight junction permeability, and SARS-CoV-2 viral load

To understand how the microbiomes of severe COVID-19 could potentially impact host immune responses, we performed Spearman correlations of the relative abundance of bacterial species identified as enriched in COVID-19 patients with plasma biomarkers

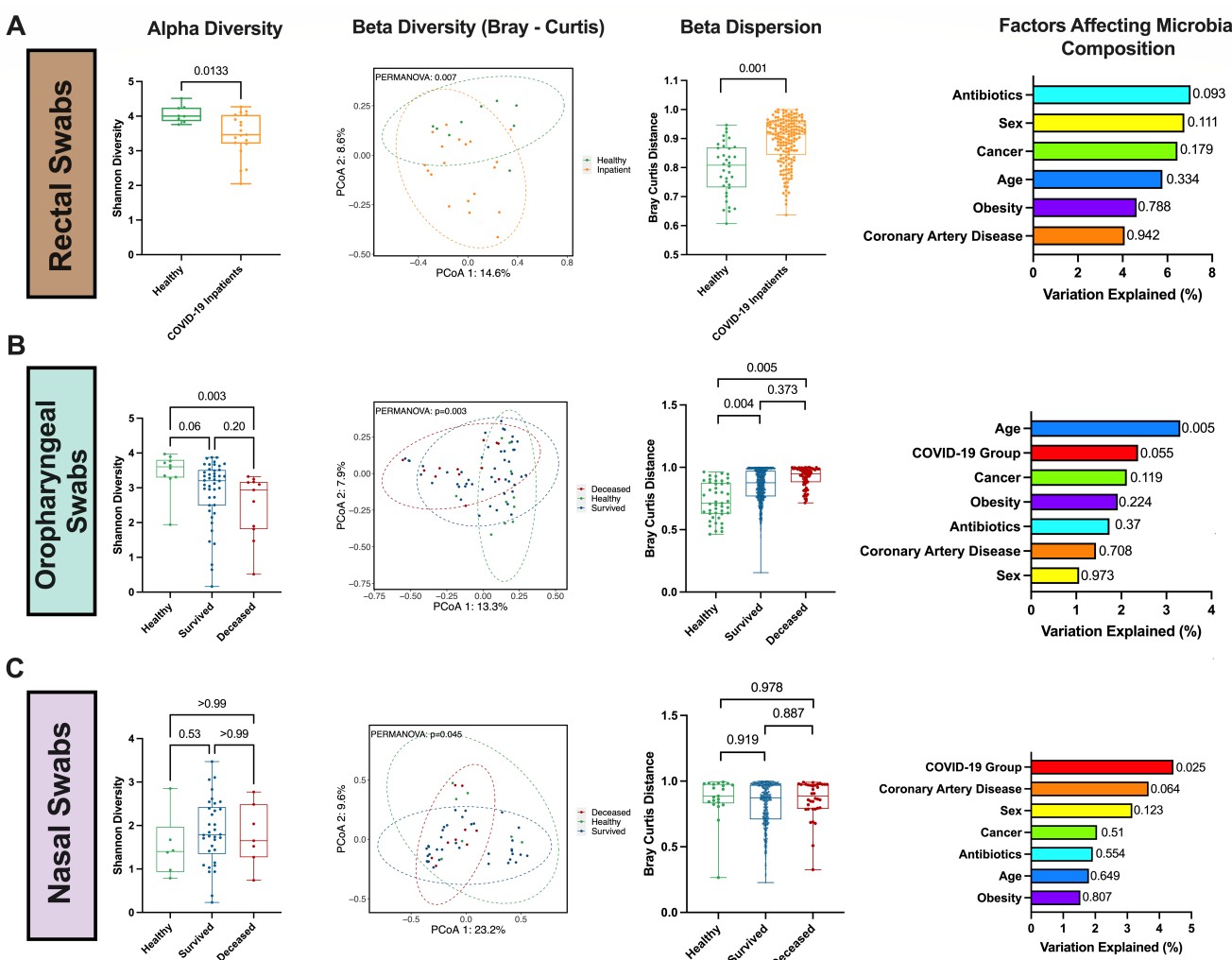

**FIG 5** Rectal, oropharyngeal, and nasal microbial communities in COVID-19 patients and healthy controls. V3V4 16S rRNA gene sequencing was used to characterize the microbial communities in healthy controls and hospitalized COVID-19 patients who survived or died in rectal swabs (A), oropharyngeal swabs (B), and nasal swabs (C). Alpha diversity was measured by the Shannon diversity index, and significance was tested by the Mann–Whitney *U* test for two group comparisons or Kruskal–Wallis tests with Dunn *post hoc* comparisons for three groups. Beta diversity was determined by the Bray–Curtis distance at the amplicon sequence variant (ASV) level and tested by PERMANOVA; ellipses represent 95% confidence intervals. Beta-dispersion was tested using the permutest function in the vegan R package with the group variances determined by the Bray–Curtis distance matrix; box plots show the Bray–Curtis distances between each sample in the same group. Factors affecting the microbial composition were determined by PERMANOVA; *P*-values for each variable are shown next to the bar plot.

that characterized severe COVID-19. In the rectal swabs, the bacterial species *Hungatella hathewayi* and *Eggerthella lenta* were enriched in the COVID-19 patients compared to healthy controls, and we observed a strong positive correlation between the relative abundance of *Hungatella hathewayi* and IL-6 plasma concentrations (*r* = 0.8, *P* = 0.003; Fig. 7A), though there were no significant correlations observed between the relative abundance of *Eggerthella lenta* and any plasma biomarkers. In the oropharyngeal swabs, *Scardovia wiggsiae* and *Veillonella parvula* were enriched in COVID-19 patients who died, and we observed significant positive correlations between the relative abundance of *Scardovia wiggsiae* with plasma concentrations of IL-6 levels (*r* = 0.39, *P* = 0.044), IL-10 (*r* = 0.45, *P* = 0.044), LBP (*r* = 0.42, *P* = 0.044), and the oropharyngeal swab SARS-CoV-2 viral load (*r* = 0.28, 0.044; Fig. 7B). These associations provide evidence for the mechanisms by which microbiome alterations in COVID-19 may drive microbial translocation and inflammation.

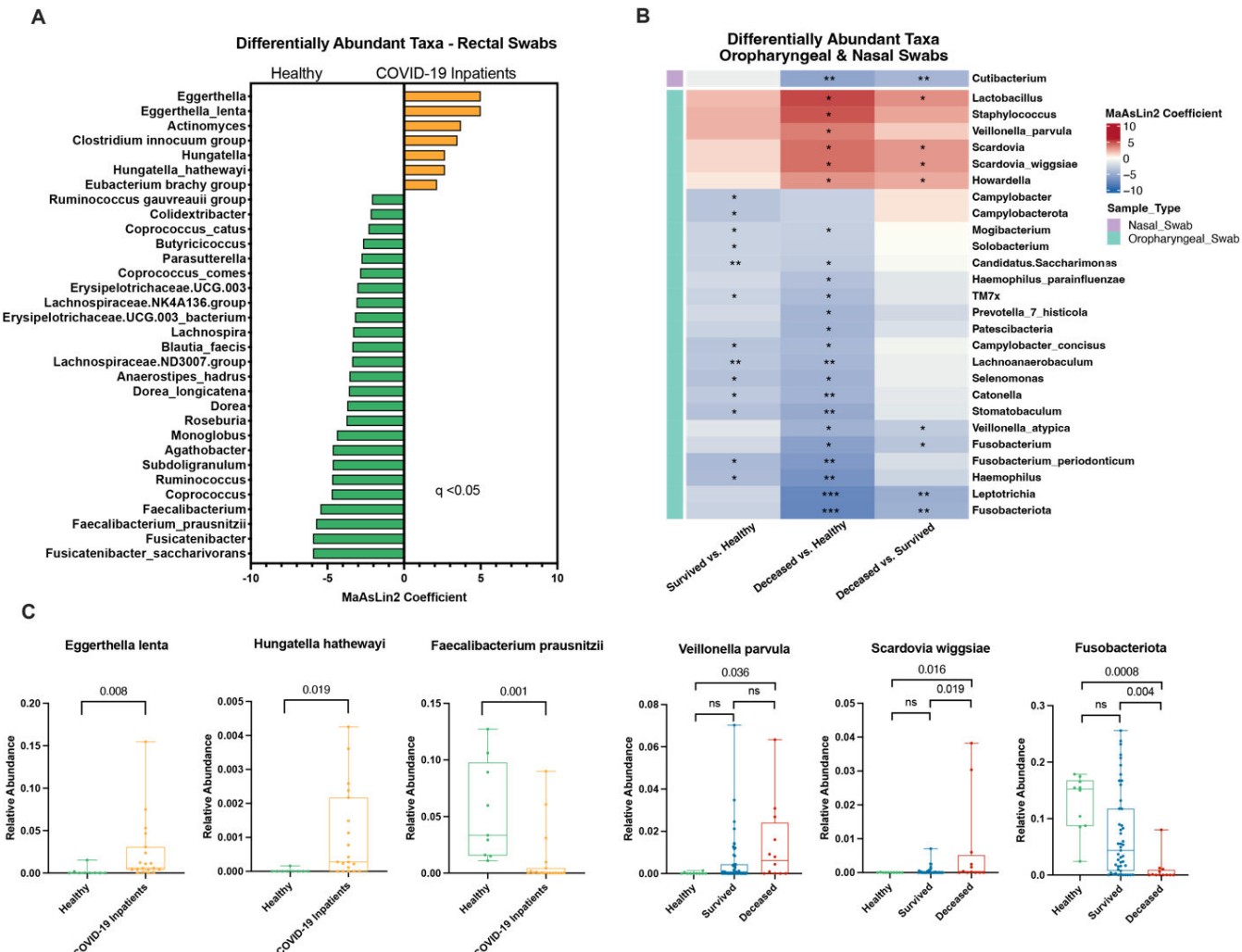

**FIG 6** Differentially abundant taxa in the rectal, oropharyngeal, and nasal swabs of patients with severe COVID-19. Differentially abundant taxa between hospitalized COVID-19 patients and healthy controls were identified using MaAsLin2 for each sample type at the phylum, genus, and species level—requiring an FDR-adjusted *P*-value less than 0.05 to be considered significant. (A) Differentially abundant taxa were identified in rectal swabs; due to a limited number of collected rectal swabs, patients from the survived and deceased groups were combined for comparison to healthy controls. (B) Differentially abundant taxa identified in oropharyngeal and nasal swabs. Red cells indicate an enrichment of that taxa in the reference group (listed first on the *x*-axis) relative to the comparison group (listed second on the *x*-axis), and blue cells indicate a depletion of that taxa. Asterisks denote FDR-adjusted *P*-values as follows: *P < 0.05, **P < 0.01, and ***P < 0.001. (C) Representative box plots showing the relative abundances of several taxa of interest from the rectal and oropharyngeal swabs.

## DISCUSSION

In this study, we examined inflammation, microbial translocation, fatty acids, hospital lab results, and the nasal, oropharyngeal, and rectal microbiomes of hospitalized COVID-19 patients and healthy controls to understand the factors that potentially contribute to COVID-19 severity. We show that the plasma concentrations of the pro-inflammatory cytokines IL-6 and IL-2 are elevated in severe COVID-19 and were the primary cytokines that distinguished COVID-19 patients who died from those who survived. Our results are consistent with several other studies with COVID-19 patients that have demonstrated higher levels of pro-inflammatory cytokines in the blood and feces (11, 57, 58) and their role in the development of a cytokine storm, a phenomenon implicated in COVID-19 severity (10, 59). In addition, IL-10, an anti-inflammatory cytokine known for its potent immunosuppressive effects (60), was dramatically elevated in COVID-19 patients when compared to healthy controls and strikingly higher in deceased COVID-19 patients. This

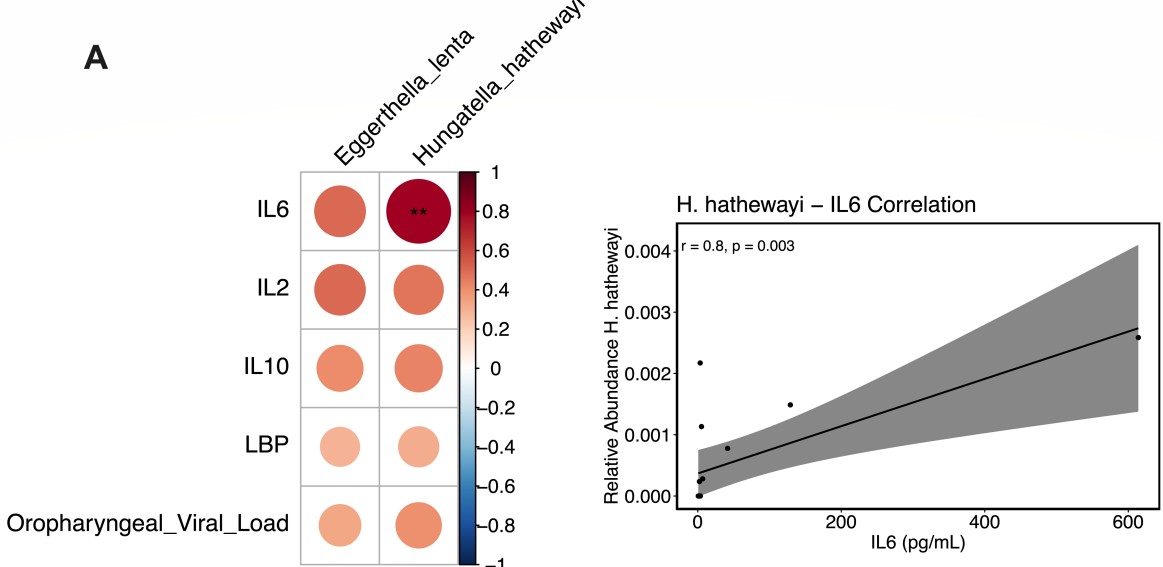

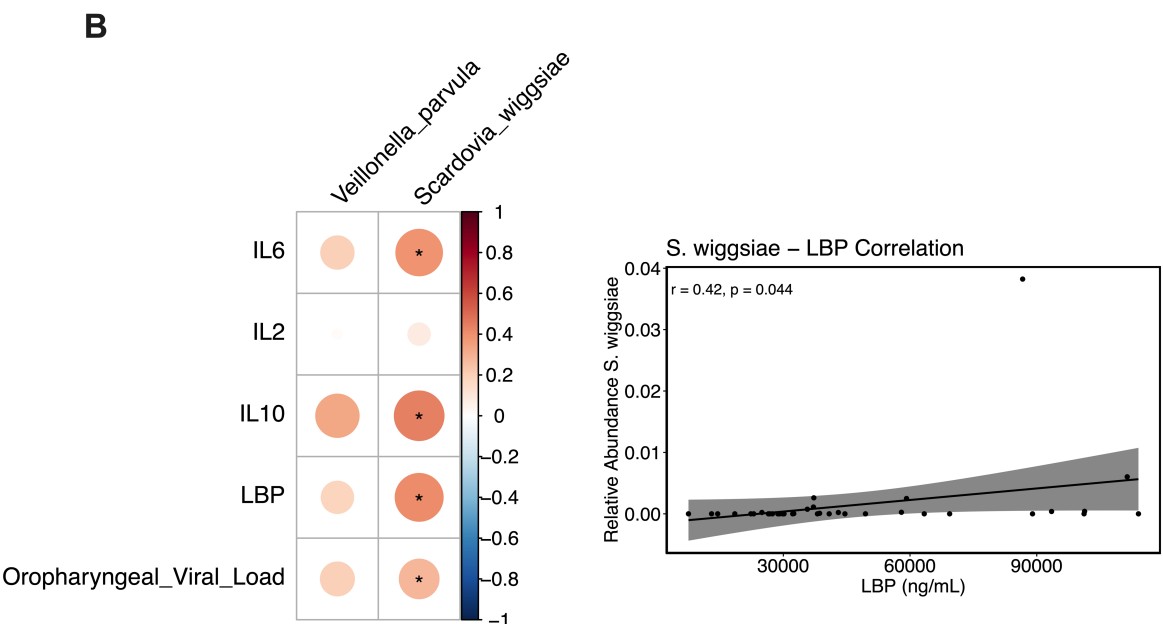

FIG 7  Bacteria enriched in severe COVID-19 are correlated with pro-inflammatory cytokines, markers of gut barrier damage, and SARS-CoV-2 viral loads. Heat maps depicting the correlation between the relative abundance of bacterial species enriched in rectal swabs (A), oropharyngeal swabs (B), and markers of inflammation, tight junction permeability, and oropharyngeal SARS-CoV-2 viral loads. Spearman's rank correlation tests were used for statistical analysis; asterisks denote FDR-adjusted *P*-values as follows: $^{*}P < 0.05$; $^{**}P < 0.01$.

finding is consistent with the results of other studies (61–63), which have demonstrated increased levels in severe COVID-19 patients. The clinical significance of the increased IL-10 levels in COVID-19 patients has been interpreted as an immune-inhibitory negative feedback tool triggered by the rapidly increasing pro-inflammatory mediators (64, 65) and an attempt to prevent tissue damage (60). Although IL-10 is conventionally known as an anti-inflammatory cytokine, the rapid early rise appears to be a distinguishing feature

of hyperinflammation during severe COVID-19 (66), and several studies have demonstrated that IL-10 levels are associated with poor outcomes in patients with COVID-19 (13, 63, 65, 67, 68). Thus, our data support the concept that inflammation drives an IL-10 response, although this immunoregulatory response does not overcome the systemic inflammation occurring in severe COVID-19 infection.

We performed a correlation analysis to further understand the relationships between plasma cytokines, gut barrier damage markers, circulating fatty acids, hospital lab results, and SARS-CoV-2 viral loads. We observed strong positive correlations between many pro-inflammatory cytokines, primarily IL-2, IL-6, TNF-α, and IFN-γ, suggesting a potential positive feedback loop as seen in cases of cytokine storm (69). We also observed positive correlations between the anti-inflammatory cytokine IL-10 and several pro-inflammatory cytokines such as IL-6, IL-2, and TNF-α, which are potentially supportive of a negative feedback loop. Furthermore, several pro-inflammatory cytokines were positively correlated with markers of gut barrier damage and intestinal permeability. Soluble immune mediators such as TNF-α and interferons produced during viral infections, including SARS-CoV-2, damage the intestinal epithelium, especially when the inflammatory response is sustained as in patients with severe COVID-19 (70–72). We observed that IL-6 was strongly correlated with LBP concentrations, suggesting that intestinal permeability and microbial translocation could be contributing to systemic immunity. Interestingly, we also observed a very strong correlation between IL-18 and the tight junction protein zonulin. IL-18 is highly expressed in mucosal sites and is important for both intestinal barrier maintenance and immunity (73). Its strong positive correlation with zonulin could possibly be indicative of an increased need for intestinal epithelial repair or an inflammatory response to microbial translocation. We also observed that albumin was negatively correlated with IL-6, IL-2, and LBP concentrations. Inflammation increases capillary permeability, which allows the escape of albumin into the interstitial space (74). Hypoalbuminemia has previously been reported as a predictive marker of COVID-19 mortality (75), and COVID-19 patients who received albumin infusions have been shown to have reduced IL-6 and IL-2R concentrations (76). Our results, therefore, may support the use of albumin in mitigating systemic inflammation from IL-6 and IL-2. Overall, these results highlight a potential signature of relationships between plasma biomarkers in severe COVID-19, as well as the relationship between systemic inflammation and intestinal permeability.

The gut microbiome harbors approximately 1,000 bacterial species (77) that have a complex relationship with an individual's health, as indicated by several recent studies demonstrating associations between the gut microbiome and allergies, inflammatory conditions, and respiratory diseases (78, 79). Furthermore, the oral microbiome is the second most diverse body site with over 700 bacterial species (80) and, along with the nasal microbiome, has been implicated in several disease states (81, 82). We observed several indicators of microbial dysbiosis present in severe COVID-19 and demonstrated altered diversity in COVID-19, indicating microbial dysbiosis. Numerous studies have described an enrichment of opportunistic pathogens associated with COVID-19 and depletion of commensal bacteria (27, 30, 46, 83). Here, we observed an enrichment of several opportunistic pathogens including *Eggerthella lenta*, *Hungatella hathewayi*, *Actinomyces, Clostridium innocuum* group, and *Eubacterium brachy* group in the rectal swabs of hospitalized COVID-19 patients compared to healthy controls, and depletion of commensal bacteria most notably including *Faecalibacterium prausnitzii, Fusicatenibacter saccharivorans, Dorea*, *Coprococcus, Ruminococcus*, and *Roseburia*. Commensal bacteria such as these have been shown to play an important role in maintaining intestinal homeostasis (84, 85), and given the markedly elevated LBP concentrations in COVID-19, their depletion along with the enrichment of opportunistic pathogens may reflect a damaged intestinal epithelium. In oropharyngeal swabs, an enrichment of *Lactobacillus*, *Staphylococcus, Veillonella parvula, Scardovia wiggsiae*, and *Howardella* and a depletion of *Leptotrichia*, *Haemophilus*, *Fusobacterium periodonticum*, *Stomatobaculum*, *Catonella*, *Selenomonas*, and *Lachnoanaerobaculum* were observed in COVID-19 patients. *Scardovia*

*wiggsiae* is a newly identified oral pathogen (86) associated with the onset of dental caries, potentially due to its ability to produce large quantities of acidic compounds such as acetic acid, and its high tolerance to fluoride (87). Its increased abundance in COVID-19 patients who did not survive suggests that dental caries and dysbiosis of the oropharyngeal microbiome may play a role in COVID-19 severity in addition to the gastrointestinal microbiome.

Our work here that explored the relationships between microbial dysbiosis, severe COVID-19, elevated inflammatory cytokines, markers of gut integrity, and microbial translocation also supports previous studies. It has been reported that an abundance of opportunistic bacteria is correlated with several inflammatory cytokines (29, 34, 78), suggesting that these could be potential targets for controlling cytokine responses in COVID-19 infection. In the present work, *Hungatella hathewayi* was positively correlated with IL-6 concentrations in plasma. Similarly, *Scardovia wiggsiae* was positively correlated with IL-6, IL-10, LBP, and SARS-CoV-2 oropharyngeal viral load. These relationships suggest that the enrichment of opportunistic pathogens in patients with COVID-19 may be contributing to systemic inflammation and disease severity, which we hypothesize is through the triggering of the production of pro-inflammatory cytokines and the disruption of epithelial tight junctions, though further studies are needed to assess precise mechanisms underlying the role of the microbiome in inflammation in the context of COVID-19. However, these data do suggest that addressing microbial dysbiosis in the gut and oral microbiomes may be targets for intervention to improve health in COVID-19 disease. Supporting this, randomized clinical trials have shown that probiotics are effective in reducing mortality or improving clinical outcomes in COVID-19 patients, though they have not been shown to be effective prophylactically in preventing COVID-19 incidence (88). These trials demonstrate the importance of the microbiome in disease severity in COVID-19. Likewise, metformin, a medication frequently prescribed for diabetes management, has been demonstrated to be advantageous for gut health and an effective treatment for COVID-19 patients (89–94). Thus, continued investigation into treatments that improve mucosal and microbial health, such as probiotics or metformin, will be critical to determining optimal treatment interventions for COVID-19. Improving the oral microbiome, either through probiotics or dental hygiene, is an underexplored treatment for COVID-19. Indeed, the bacteria or biomarkers identified in this study could be investigated for their ability to predict the clinical outcomes in newly infected patients as well as personalize treatments to improve outcomes in individuals with COVID-19 disease.

Due to the extremely stressful hospital environment early in the COVID-19 pandemic, together with the voluntary nature of how samples were collected, and the isolation procedures in place at the hospital at the time of collection, several sample types were collected in limited numbers. For example, only two rectal swabs could be collected from deceased COVID-19 patients making it statistically impossible to compare the rectal microbial community between healthy controls, patients who survived, and patients who died. We were also limited by the number of healthy controls enrolled in the study with a total of 12, but only five plasma samples were collected given the lack of procedures performed with healthy adults during the initial COVID-19 lockdown. Furthermore, we lacked a non-COVID hospitalized control, which may have shown differences between COVID-19-infected patients and a similarly hospitalized pneumonia patient. Lastly, due to limited sample sizes and available metadata for our healthy controls, we did not adjust for potential confounding variables such as age, sex, BMI, or antibiotic usage in this study. This is an unfortunate limitation that occurred due to the worldwide COVID-19 pandemic and associated mandatory lockdowns, which made it difficult to collect healthy samples and associated metadata. Future studies will be needed that fully address confounding variables associated with severe COVID-19 as these factors have been shown to impact cytokine production and microbial composition (95, 96). We did investigate the effect of several of these factors in the overall microbial composition of the hospitalized COVID-19 patients and did not find that they

were large influences, apart from the effect of age on the oropharyngeal microbiome. However, despite these caveats, given the extreme conditions in which samples were collected in the early pandemic in 2020, these results are highly robust and indicate several mechanisms that underlie COVID-19 pathogenesis.

In summary, our data suggest that severe COVID-19 is associated with markers of disrupted gut barrier integrity and microbial translocation, pro- and anti- inflammatory cytokines, altered circulating fatty acids, and disruptions in the microbiome of rectal, oropharyngeal, and nasal swabs. Our study sheds light on the critical role of microbial dysbiosis and the resultant microbial translocation in the pathophysiology of severe COVID-19 and the potential relationship between specific microbes and systemic inflammation. By understanding these underpinnings of COVID-19, this work may help to identify biomarkers for risk classification and build a foundation for developing novel strategies to prevent or reduce the severity of COVID-19 and improve the management of COVID-19 patients.

## MATERIALS AND METHODS

### Study design

In this cross-sectional observational study, we enrolled 86 patients hospitalized with COVID-19 from the M Health Fairview Bethesda Hospital and 12 noninfected controls between May and October 2020 in Minnesota, United States. All participants had the option to donate whole blood, nasal swabs, rectal swabs, and oropharyngeal swabs. Due to the voluntary nature of sample collection, the number of available samples varies for each sample type, as summarized in Table S1. For example, although 12 healthy controls were enrolled, only five plasma samples were able to be collected for this group. By the end of study enrollment, 17 COVID-19-infected patients had died and 69 survived; we define those in the deceased group as having more severe illness than those who survived. Demographics, prior medical history, hospital lab results, and hospitalization information were obtained from the electronic medical record, as shown in Table 1.

### Cytokine and gut barrier damage measurements

Whole blood samples were collected in ethylenediaminetetraacetic acid (EDTA) anticoagulant tubes, and plasma was isolated by centrifugation and stored at −80℃. Cytokine concentrations were measured in plasma using the ProteinSimple SimplePlex assay (IL-1β, IL-2, IL-6, IL-10, IL-12p70, IL-18, TNF-α, and IFN-γ) and the Milliplex High Sensitivity T Cell Panel (IL-8, IL-17A, and IL-23). Gut barrier damage biomarkers, including LBP (cell sciences CKH113), sCD14 (R&D Systems QK383), zonulin (MyBioSource MBS706368), and I-FABP (R&D Systems DFBP20), were measured using commercially available assays according to the manufacturer's guidelines. Undetected values were replaced with each assay's respective limit of detection.

### Measurement of circulating fatty acids in plasma

Samples were processed using a modified previously described protocol (97). Briefly, 90 µL of EDTA plasma was combined with 10 µL of an internal standard mix [2.5 µL of each 10 mg/mL deuterated standard, acetonitrile/water (1/1; vol/vol) to a final volume of 1,000 µL]. Subsequently, samples were combined with 420 µL of cold methanol, vortexed for 10 seconds, incubated for 10 min at −20℃, and centrifuged at room temperature for 10 min at 14,000 × $g$. Forty microliters of supernatant was transferred to a new 2-mL centrifuge tube and combined with 20 µL of 3-nitrophenylhydrazine and 20 µL of 1-ethyl-3-dimethylaminopropyl carbodiimide solution. Samples were incubated at 40℃ for 30 min and then diluted with 1,520 µL water/acetonitrile (9/1; vol/vol). LC-MS/MS analysis was carried out utilizing a multiple reaction monitoring method for analyte detection on an Agilent LC-1290 Infinity II (Agilent Technologies, Consumer Electronics Inc, Santa Clara, CA) coupled with an Agilent 6495C mass spectrometer (Agilent

Technologies, Consumer Electronics Inc, Santa Clara, CA). Chromatographic separation was achieved through the utilization of a C18 Acquity UPLC BEH column (2.1 × 100 mm, 1.7 µm) at 60℃. Mobile phases of separation consisted of A: methanol/2-propanol (1/1; vol/vol) and B: methanol/water (1/1; vol/vol). The negative ion mode was utilized for acquisition, with an injection volume of 10 µL. Flow rates began at 0.5 mL/min for 0–4 min, with a subsequent decrease to 0.25 mL/min for 4–12 min, before returning to the initial flow rate of 0.5 mL/min for the remainder of the run time, 12–25 min. The elution began with a linear gradient of 95%–55% A, 0–4 min; 4–12 min linear gradient 55%–52.5% A; 12–15 min linear gradient 52.5%–2% A; 15–17 min isocratic hold 2% A; 17–17.1 min linear gradient 2%–95% A; 17.1–25 min isocratic hold 95% A. The acquired data were imported into Agilent MassHunter software for additional analysis. Undetected values were replaced with the lowest interpolated value in the sample set.

## DNA/RNA extractions, viral load measurements, and 16S metagenomic sequencing

DNA and RNA were extracted from nasal, oropharyngeal, and rectal swabs using the Qiagen AllPrep PowerViral DNA/RNA extraction kit according to manufacturer instructions. Bacterial template DNA quantity was determined by performing qPCR using the V3_357F_Nextera and V4_806R_Nextera primers targeting the V3V4 locus of the rRNA gene. Samples were diluted with water if necessary, and amplicons were generated using a 25-cycle PCR reaction using the same V3V4 primers. PCR products were diluted 1:100 in water and subjected to a second 10-cycle PCR reaction to attach Illumina sequencing primer-compatible DNA regions as well as individual barcodes for each sample. As previously described, the samples were all uniquely dual-indexed (98). Final PCR products were normalized using SequalPrep kits (Invitrogen), pooled into sequencing libraries, and cleaned with AMPure XP mag beads (Beckman Coulter). Sequencing libraries were loaded onto an Illumina MiSeq using a 2 × 300 v3 flow cell (Illumina, San Diego, CA) and sequenced to an average depth of 87,099 reads per sample. Droplet digital PCR (ddPCR) of the N1 and N2 genes was used to determine SARS-CoV-2 viral loads.

## Hospital lab results

Hospital lab results for albumin, AST, ALT, LDH, neutrophil count, and creatinine kinase were extracted from the electronic medical record from admission to discharge. Lab results within ±3 days from when the patient donated samples were averaged for downstream analysis.

## Data processing

Variable region primers were removed from the demultiplexed sequences using cutadapt (version 4.4) and then quality-filtered, trimmed, denoised, and merged in R (version 4.2.1) using the dada2 package (99, 100). Merged sequences were further filtered to only include those within the expected base-pair length of the V3V4 amplicon and then used to generate an amplicon sequence variant (ASV) table. Taxonomy was assigned using the SILVA v138 database (47, 101). Samples with less than 1,000 sequencing reads or less than 100 16S copies/µL were removed from the analysis. Unassigned ASVs or those assigned to Archaea, Eukaryota, Mitochondria, or Chloroplast or prevalent in less than 5% of samples were removed.

## Microbiome analysis

For alpha diversity, samples were first normalized by scaling with ranked subsampling (SRS) (version 0.2.3) using the SRS.shiny.app for the determination of Cmin. The Shannon diversity index was measured at the ASV level using the microbiome package (version 1.20.0), differences between two groups were determined by Mann–Whitney *U*

tests, and differences between three groups were determined by Kruskal–Wallis tests followed by *post hoc* Dunn tests in GraphPad Prism (version 9.5.1). Beta diversity was determined by multidimensional scaling of the Bray–Curtis distance on the relative abundance of the ASV table and tested by PERMANOVA using the Adonis2 function from the vegan package (version 2.6-4). Beta-dispersion was tested using the permutest function in the vegan R package with the group variances determined by the Bray–Curtis distance matrix. Differential abundance of taxa was determined using the MaAsLin2 package (version 1.12.0) at the phylum, genus, and species levels requiring a minimum prevalence of 30%. Spearman correlations between the relative abundance of differentially abundant species and biomarkers that were elevated in deceased patients were performed using the psych package (version 2.3.3) and visualized using the corrplot package (version 0.92). All *P*-values were adjusted for false discovery rate using the Benjamini–Hochberg method (48) with a significance level of 0.05. Comparisons for nasal, oropharyngeal, and rectal swabs were adjusted separately.

## Statistical analysis

Statistical analysis was performed using R (version 4.2.3). The distribution of each continuous variable was visualized to determine whether it was normally or nonnormally distributed. Comparisons with more than two groups were performed using Kruskal–Wallis tests followed by pairwise Dunn tests if significant. Comparisons between two groups were done using two-sided Welch or Wilcox tests for normally or nonnormally distributed variables. Categorical variables were compared using chi-squared or Fisher exact tests. Correlations between cytokines, gut barrier damage markers, SCFAs, and hospital lab results were determined using Spearman correlations. Supervised clustering was performed by sPLS-DA using the MixOmics R package (version 6.22.0). All *P*-values were appropriately adjusted using the Benjamini–Hochberg method (48).

## ACKNOWLEDGMENTS

The authors thank the University of Minnesota Genomics Core for performing 16S sequencing and ddPCR, the University of Minnesota Cytokine Reference Laboratory for performing cytokine concentration measurements, and the University of Minnesota Clinical and Translational Institute for statistical support. The authors acknowledge Nichole R. Klatt's Professorship and the University of Minnesota Department of Surgery.

Conceptualization was performed by C.M.B., L.S., and N.R.K.; supervision was performed by N.R.K., T.W.S., C.J.T., P.J.S., C.T.B., T.D.B., and S.E.S.; funding acquisition was performed by N.R.K.; investigation was performed by C.M.B., J.S., R.C., C.A.B., C.R.G., A.V., T.S., and Z.J.M.; formal analysis was performed by C.M.B.; visualization was performed by C.M.B.; writing (original draft) was performed by C.M.B. and R.L.; writing (review and editing) was performed by N.R.K., M.B., and L.S.

## AUTHOR AFFILIATIONS

[1]Department of Surgery, University of Minnesota, Minneapolis, Minnesota, USA

[2]Department of Biochemistry, Molecular Biology and Biophysics, University of Minnesota, Minnesota, Minneapolis, USA

[3]College of Biological Sciences, University of Minnesota, Minnesota, Minneapolis, USA

[4]Department of Medicine, University of Minnesota, Minnesota, Minneapolis, USA

[5]Masonic Cancer Center, University of Minnesota, Minnesota, Minneapolis, USA

[6]Department of Microbiology and Immunology, University of Minnesota, Minnesota, Minneapolis, USA

[7]Department of Biostatistics and Health Data Science, University of Minnesota, Minnesota, Minneapolis, USA

[8]National Cancer Institute, Center for Cancer Research, Vaccine Branch, Animal Models and Retroviral Vaccines Section, National Institutes of Health, Bethesda, Maryland, USA

## AUTHOR ORCIDs

Christopher M. Basting ⓘ http://orcid.org/0009-0001-7744-3673
Nichole R. Klatt ⓘ http://orcid.org/0000-0003-2968-5480

## AUTHOR CONTRIBUTIONS

Christopher M. Basting, Conceptualization, Formal analysis, Investigation, Visualization, Writing – original draft | Robert Langat, Writing – original draft | Courtney A. Broedlow, Investigation | Candace R. Guerrero, Investigation | Tyler D. Bold, Supervision | Melisa Bailey, Writing – review and editing | Adrian Velez, Investigation | Ty Schroeder, Investigation | Jonah Short-Miller, Investigation | Ross Cromarty, Investigation | Zachary J. Mayer, Investigation | Peter J. Southern, Supervision | Timothy W. Schacker, Supervision | Sandra E. Safo, Supervision | Carolyn T. Bramante, Supervision | Christopher J. Tignanelli, Supervision | Luca Schifanella, Conceptualization, Writing – review and editing | Nichole R. Klatt, Conceptualization, Funding acquisition, Supervision, Writing – review and editing

## DATA AVAILABILITY

All 16S rRNA sequencing data are available under NCBI BioProject PRJNA1065883. All other data and codes are available upon request.

## ETHICS APPROVAL

Informed consent was collected from each participant, and the University of Minnesota Institutional Review Board approved the study.

## ADDITIONAL FILES

The following material is available online.

### Supplemental Material

**Figure S1 and Table S1 (Spectrum00680-24-s0001.pdf).** Fig. S1: Hospitalization of COVID-19 patients and disease outcome.
Table S1: Samples available for each group.

### Open Peer Review

**PEER REVIEW HISTORY (review-history.pdf).** An accounting of the reviewer comments and feedback.

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
