## [Reviewer comments · Microbiology Spectrum]

Microbiology Spectrum

SARS-CoV-2 infection is associated with intestinal permeability, systemic inflammation and microbial dysbiosis in hospitalized patients

Christopher Basting, Robert Langat, Courtney Broedlow, Candace Guerrero, Tyler Bold, Melisa Bailey, Adrian Velez, Ty Schroeder, Jonah Short-Miller, Ross Cromarty, Zachary Mayer, Peter Southern, Timothy Schacker, Sandra Safo, Carolyn Bramante, Christopher Tignanelli, Luca Schifanella, and Nichole Klatt

Corresponding Author(s): Nichole Klatt, University of Minnesota Twin Cities

Review Timeline:

Submission Date:	May 22, 2024
Editorial Decision:	August 12, 2024
Revision Received:	August 26, 2024
Accepted:	September 3, 2024

Editor: Steven Frese

Reviewer(s): The reviewers have opted to remain anonymous.

Transaction Report:

DOI: <https://doi.org/10.1128/spectrum.00680-24>

Re: Spectrum00680-24 (SARS-CoV-2 infection is associated with intestinal permeability, systemic inflammation and microbial dysbiosis in hospitalized patients)

Dear Dr. Nichole R Klatt:

Thank you for the privilege of reviewing your work. Below you will find my comments, instructions from the Spectrum editorial office, and the reviewer comments.

Revision Guidelines

Sincerely,
Steven Frese
Editor
Microbiology Spectrum

Reviewer #1 (Comments for the Author):

Review for Microbiology Spectrum:

Overall, this was a fascinating study completed at one time point. I think defining disease severity will help inform the impact of their work. As a new microbiome researcher their methods appear sound although I am new to the droplet PCR technique for assessing viral loads. Below are overall comments and some line by line thoughts:

One of the interesting pieces of the manuscript the author continuously highlights is the severity of COVID-19 and even mentions this in the abstract. The wording leaves me to believe there is a questionnaire or clinician-based screening for the

severity of the disease, but I did not see one in the research. Could the authors either reword this disease "severity" throughout the paper or define disease severity (e.g. perhaps hospitalized vs. deceased)? An example of this is page 9 line 203, can you say that "several markers increase with disease severity" given that the only participants in the study are hospitalized? Once this is completed it will be easier to understand authors interpretation of data and overall conclusions.

Introduction: Write out MCFA acronym before using acronym alone.

Methods:

Page 11: L242: define mucosal dysfunction (rhinitis?)

Was there any significance on page 13 and 14? (Figure 6B) Can you confirm if all or some of the bacteria listed here were significant?

Discussion:

Page 18: L403: say more on Scardovia -> Why dysbiosis does it indicate that dysbiosis is present? Is that because it is a new bacteria?

Page 19: Line 420: Metformin? You introduced a new topic and I'm not entirely certain how this relates could you pull together your thought process or what metformin does to treat COVID-19? I assume it is used to reduce inflammation based on your study. It may not be necessary to add these two lines at all.

Methods: Were there 5 or 12 Healthy Controls? Text and table 2 were different. Defined in discussion, but could you define earlier near table 2 in the text?

Figure 1 and 2: Why do some plots say Kruskal Wallis and not others? Consider just using the p-value as to not confuse the reader into thinking different tests were completed for different immune markers.

Reviewer #2 (Comments for the Author):

The main content of this article is the study of the association between SARS-CoV-2 infection and intestinal permeability, systemic inflammation, and microbial dysbiosis in hospitalized patients. The study involved various sample types such as plasma, rectal swabs, oropharyngeal swabs, and nasal swabs, providing multi-angle data for a comprehensive understanding of the pathophysiology of COVID-19.

Major comments:

1. Potential confounding variables were not adjusted: The study did not adjust for potential confounding variables such as age, gender, especially antibiotic use (Fig.5). The direction for improvement is to include these variables in the data analysis to control for their potential impact.
2. The authors found that pro-inflammatory cytokines could distinguish between deceased and surviving patients. It raises curiosity about how the microbiome or the combined use of cytokines and microbiome might yield results.
3. Fig.2 How were these SCFAs and markers determined to be selected as evaluative indicators?

Minor comments:

1. Some of the text in the Figures is too small: Fig.6 A-B

Response to Reviewers: Spectrum00680-24

We thank the reviewers for their careful review and thoughtful insights on our manuscript. We feel we have fully addressed all reviewer concerns and have improved the manuscript, we hope it is now suitable for publication in Spectrum.

Reviewer #1:

- **Could the authors either reword this disease "severity" throughout the paper or define disease severity (e.g. perhaps hospitalized vs. deceased)?**
 - Severity of COVID-19 was defined as whether the patient had survived or died by the end of study enrollment, and groups were referred to as "Survived" or "Deceased". We unfortunately did not have a questionnaire or clinician-based screening for defining COVID-19 severity as part of this study, but that could be a future improvement. **We have added additional clarification in the results and methods sections defining these groups.**
- **Write out MCFA acronym before using acronym alone.**
 - Medium-chain fatty acids (MCFAs) is now written out just before its first use as an acronym (line 118)
- **Page 11: L242: define mucosal dysfunction (rhinitis?)**
 - "Mucosal dysfunction" in this sentence was meant to highlight the positive correlations seen between SARS-CoV-2 viral loads, pro-inflammatory cytokines (IL-6), and markers of microbial translocation (LBP/sCD14) or gut barrier damage (zonulin). So here, "mucosal dysfunction" primarily refers to the increased microbial translocation and gut barrier damage that is associated with SARS-CoV-2 viral loads and pro-inflammatory cytokines. **We have added additional clarification.**
- **Was there any significance on page 13 and 14? (Figure 6B) Can you confirm if all or some of the bacteria listed here were significant?**
 - Yes, some of the bacteria shown in Figure 6B are significant. This is shown by the asterisks within each cell of the heatmap and defined in the figure legend. **We have increased the size of the asterisks to better highlight this.**
- **Page 18: L403: say more on Scardovia -> Why dysbiosis does it indicate that dysbiosis is present? Is that because it is a new bacteria?**
 - **We have added additional text to go in depth further on this.** *Scardovia wiggsia* has recently been described as a pathogenic oral bacterium that is associated with caries (tooth decay) and has been shown to have a high capacity for producing acidic compounds (mainly acetic acid) as a potential mechanism for causing caries. An increased abundance of this pathogenic bacterium in the oral cavity therefore is suggestive of dysbiosis because it could contribute to caries and inflammation, which we show with a positive correlation between its abundance and IL-6/IL-10/LBP.
- **Page 19: Line 420: Metformin? You introduced a new topic and I'm not entirely certain how this relates could you pull together your thought process or what**

metformin does to treat COVID-19? I assume it is used to reduce inflammation based on your study. It may not be necessary to add these two lines at all.

- Our main point here is to highlight that treatments focused on improving microbial dysbiosis (and reducing inflammation), such as Metformin, may be effective at also improving COVID-19 clinical outcomes. We have added in additional information to make this point more clear, also including new research on the effective use of probiotics on COVID-19, further exemplifying the role of the microbiome in COVID-19 disease.
- Were there 5 or 12 Healthy Controls? Text and table 2 were different. Defined in discussion, but could you define earlier near table 2 in the text?
 - We enrolled 12 healthy controls, however samples were collected on a voluntary basis, so for some sample types (e.g. plasma) we only had 5 healthy control samples available. We have added a table (Supplemental Table 1) which highlights the number of each sample type available for each group (Healthy, Survived, Deceased) and added additional text in the Study design section of the methods describing this.
- Figure 1 and 2: Why do some plots say Kruskal Wallis and not others? Consider just using the p-value as to not confuse the reader into thinking different tests were completed for different immune markers.
 - Pairwise comparisons using Dunn's test were only performed if the Kruskal-Wallis test was significant (FDR p-value < 0.05) in order to prevent performing unneeded statistical testing. We have made this more clear in the figure legend.

Reviewer #2:

- Potential confounding variables were not adjusted: The study did not adjust for potential confounding variables such as age, gender, especially antibiotic use (Fig.5). The direction for improvement is to include these variables in the data analysis to control for their potential impact.
 - This is an unfortunate limitation of the study that we did during a trying time of the worldwide COVID-19 pandemic and mandatory shut downs. We lacked the metadata for the confounding variables for our healthy controls. For the microbiome results, we do show the influence of some of these confounding variables on the COVID-19 patients (Fig 5 "Factors Affecting Microbial Composition"). We did not see a significant impact of antibiotic use on the overall microbial composition in any of the swabs. Age had the highest influence of any of the confounding variables, but only in the oropharyngeal swabs. The limitation of this is that this only includes the hospitalized COVID-19 groupings (Survived/Deceased), not the healthy controls. We have added additional text highlighting this limitation.

- The authors found that pro-inflammatory cytokines could distinguish between deceased and surviving patients. It raises curiosity about how the microbiome or the combined use of cytokines and microbiome might yield results.
 - This is a good point which we are exploring in follow up studies. We have added some text regarding this in the discussion.
- Fig.2 How were these SCFAs and markers determined to be selected as evaluative indicators?
 - The SCFAs propionate, butyrate and isovaleric acid are some of the primary metabolites produced by the gut microbiome. LBP and sCD14 are commonly used biomarkers for microbial translocation, as is I-FABP for enterocyte damage, and zonulin for tight junction function in the gut. The medium chain fatty-acids including decanoic acid and nonanoic acid were primarily chosen because they are more abundant in human plasma and have not been thoroughly evaluated in COVID-19 patient cohorts
- Some of the text in the Figures is too small: Fig.6 A-B
 - We've increased the size of the text in these figures, thank you for pointing this out.

Re: Spectrum00680-24R1 (SARS-CoV-2 infection is associated with intestinal permeability, systemic inflammation and microbial dysbiosis in hospitalized patients)

Dear Dr. Nichole R Klatt:

Your manuscript has been accepted, and I am forwarding it to the ASM production staff for publication. Your paper will first be checked to make sure all elements meet the technical requirements. ASM staff will contact you if anything needs to be revised before copyediting and production can begin. Otherwise, you will be notified when your proofs are ready to be viewed.

Sincerely,
Steven Frese
Editor
Microbiology Spectrum